# Aerial Tramway Sustainable Monitoring with an Outdoor Low-Cost Efficient Wireless Intelligent Sensor

**Rafael Cardona Huerta** [1] **, Fernando Moreu** [2,*] **and Jose Antonio Lozano Galant** [3]

1   Center for Intelligent Infrastructure, Research Engineer, 202 Engineering Research Laboratory, Department of Civil Engineering, University of Missouri Science and Technology, 500 W 16th Street, Rolla, MO 65409, USA; rafaelcardonahuerta@mst.edu

2   Centennial Engineering Center 3056, Department of Civil Engineering, University of New Mexico, MSC01 1070, Albuquerque, NM 87131, USA

3   Civil Engineering, Faculty of Civil Engineering of Ciudad Real, University of Castilla-La Mancha, Ed. Politecnico Camilo Jose Cela, 13071 Ciudad Real, Spain; joseantonio.lozano@uclm.es

*   Correspondence: fmoreu@unm.edu; Tel.: +1-505-277-1784

**Abstract:** Infrastructures such as aerial tramways carry unique traffic operations and have specific maintenance requirements that demand constant attention. It is common that old structures lack any type of automatization or monitoring systems, relying only on human judgment. Owners are interested in implementing techniques that assist them in making maintenance decisions, but are reluctant to invest in expensive and complex technology. In this study, researchers discussed with the owners different options and proposed a sustainable and cost-efficient solution to monitor the Sandia Peak Tramway operations with just two strategically located acceleration sensors. To maximize the success options researchers worked with the owners and developed a sensor that satisfied their needs. A Low-cost Efficient Wireless Intelligent Sensor 4—Outdoors (LEWIS 4) was developed, tested and validated during the experiment. Two solar-powered units were installed by the tramway staff and recorded data for three days. When retrieved, researchers analyzed the data recorded and concluded that with only two sensors, the acceleration data collected were sufficient to determine the position and location of the tramway cars. It was also found that the sensor on the tower provides data about the cable–tower interaction and the forces caused by the friction on the system, this being a critical maintenance factor. This work summarizes a methodology for infrastructure owners consisting of guidelines to design a sustainable and affordable monitoring approach that is based on the design, development and installation of low-cost sensors.

**Keywords:** Arduino; aerial tramway; transportation; monitoring; acceleration





## 1. Introduction

Civil infrastructure maintenance guarantees safe operations. In general, maintenance operations depend on owners either estimating the load frequency and weight, or manually counting them. Developing new technologies to monitor structures and improve maintenance activities is a new active field of research [1–4]. The concept of monitoring structures to assess their health condition is approximately 30 years old [5]. More recently, researchers have focused their monitoring efforts to inform maintenance actions and engineers monitor the loads in transportation infrastructures to inform owners of management operations [6–8] resulting in improvements in environmental and operational efficiency [9,10]. Owners can use this loading information to inform a consequence-based maintenance program that translates into a sustainable use of the resources to carry out those maintenance activities [11,12]. Monitoring the frequency of loads using structural responses helps owners to change and update maintenance programs using objective data.

Researchers are developing new techniques to assist maintenance activities placing wireless smart sensors (WSSs) on key points of the structure to obtain information about

its behavior [13,14]. Despite the development of necessary technology [15] and field applications in multiple academic cases [16], these methods are not always installed by owners due to both their complexity [17] and high cost [18]. As a result, the application of structural monitoring with WSSs is still transitioning from specific applications [19] to widespread implementation by all infrastructure owners. Making WSS monitoring technologies more accessible and extending their benefits to a broader range of applications is a growing research field [20]. To overcome the challenge of implementation researchers have been working on low-cost sensors to standardize infrastructure monitoring through WSSs [21–23]. In this study, researchers tested their monitoring capabilities on real structures to demonstrate their value to infrastructure managers.

WSS field implementations need to rely on portable power sources [24]. Commonly, sensors do not have a source of energy and depend on batteries. Therefore, batteries are crucial when monitoring for extended periods of time. Long-term deployment capability is essential to obtain responses to dynamic load events through time. Researchers have developed solar energy harvesting solutions for WSS systems and minimized the energy consumption through software enhancements [25]. Development of affordable and reliable long-term WSSs is a challenge for the implementation of large WSS networks in the field.

Aerial tramways and chair lifts are not only a tourist attraction but an alternative mode of transportation in situations where conventional public transport modes are not feasible [26,27]. Depending on the age of the structure, the grade of automation varies from fully manual operated systems to completely automatic operating [28]. However, these systems are developed by external entities or are privately owned and operated, and owners tend to develop their specific in-house solutions for long term management. It would be valuable to develop simple monitoring systems that owners can understand and implement for their operations [29]. If the owners participate in the development of the low-cost sensing solution, they can inform how to use it in the future for operations interesting to them during the sensor development. This can be a good symbiotic collaboration for industry and research, as once the owners are more involved in the sensor development, they will also provide inputs and critical suggestions on how they could be improved for further development, which benefits research growth and implementation of new solutions.

In this study, researchers designed and built a low-cost, long-term deployable WSS called Low-cost Efficient Wireless Intelligent Sensor LEWIS 4. LEWIS 4 is the latest generation of the LEWIS series [21–23,30,31]. LEWIS 4 is a sustainable self-powered sensor that uses a solar energy harvesting power supply system. The design of this energy system makes the sensor autonomous, allowing it to record over days. Comparable to other acceleration sensors, LEWIS 4 can collect continuous acceleration data, which adds to the low cost, easy to build and quick to deploy philosophy of LEWIS. In this research two LEWIS 4 WSSs were installed on the Sandia Peak Aerial Tramway in Albuquerque, New Mexico. The data collected provides information for the amount of flights, time, speed and position that is currently collected manually. Researchers also identified, by discussing the sensor development with the Tramway owners, that these data can inform the owners of mechanical properties. As a result, this paper shows how low-cost sensor development with owners can benefit the advancement of automatic monitoring of infrastructure.

LEWIS 4 was designed to inform owners of transportation operations in the context of the stakeholder prioritization of maintenance and management. The main innovation of this new generation of LEWIS is the long-term capability and autonomous data acquisition. The main purpose is to collect and save data to inform the Tramway managers of transportation activity related to maintenance and management currently collected visually. The potential of the LEWIS series is its versatility to, using the same basis, adapt to different purposes such as training, laboratory experiments or field-sensing. LEWIS 4 has advanced previous work adapting the system for long-term outdoor monitoring applications. Table 1 shows the evolution of the sensors and compares its capabilities. LEWIS 4 has advanced previous work adapting the system for long-term outdoor monitoring applications. Researchers tested the LEWIS 4 for several days in order to test its

ability to collect acceleration data autonomously. It is also of interest that LEWIS 4 was developed to resist extreme conditions for deployment, operations and retrieval unique to the Tramway of Albuquerque, which is a preferred feature by owners and stakeholders in terms of using low-cost sensors [32]. This research, in collaboration with the owners of the tramway infrastructure, develops, tests and deploys a smart sensor that meets the requirements to successfully monitor the traffic operations of the infrastructure.

**Table 1.** LEWIS comparison. LEWIS 1 [21]; LEWIS 2 [23,30]; LEWIS 3 [21,23].

| SENSOR | LEWIS 1 | LEWIS 2 | LEWIS 3 | LEWIS 4 |
|---|---|---|---|---|
| Wireless | No | Yes | Yes | Yes |
| Power Source | Wire | Battery | Battery-Solar panel | Battery-Solar panel |
| Ideal deployment situations | • Training and education<br>• Easy access placements<br>• Minutes duration deployment<br>• No angular displacement | • High accuracy requirement<br>• Easy access placements<br>• Minutes duration deployment<br>• Angular displacement | • High accuracy requirement<br>• Easy access placements<br>• Hours duration deployment<br>• Angular displacement | • High accuracy requirement<br>• Difficult access placements<br>• Days duration deployment<br>• Angular displacement |
| Acceleration sensor | MPU 6050 | MPU 9250 | MPU 9250 | MPU 9250 |
| Acceleration range (g) | $\pm 2$ | $\pm 16$ | $\pm 16$ | $\pm 16$ |
| Frequency (Hz) | 100 | 500 | 500 | 250 |
| Construction Price ($) | 65 | 100 | 250 | 250 |

## 2. Field Requirement Analysis and Methodology for Sensor Development

We focused our efforts on determining the sensor requirements and designing the experiment plan to collect enough data to let us monitor the structure. Initially, we followed a plan to develop and test a sensor that would work keeping the field deployment in mind. The problems to address were as follows: to determine the ideal deployment duration, the hardware and software best design, solve the power supply problem, plan how to manage the data and finally test the device to make sure that it was ready to deploy. At the same time we worked on the tramway to decide what would be the minimum number of sensors that would be needed to collect enough information. The strategy followed was to deploy two sensors that, located on one of the two towers and one of the two cars, could collect acceleration data for at least three days. The sensor´s accuracy and reliability were tested prior to the field deployment on the tramway in an experiment where researchers studied the data acquisition and power system of the sensor.

The hypothesis of this work is that engaging the owner before the design and fabrication of sensors identifies which data can be of value from the owner's perspective early in the monitoring. According to owner, the main problems creating barriers for implementation for field monitoring of existing structures are the cost and time required to obtain early results that can inform about the value of the data [15]. Even when owners design the sensing system, they are in general not involved in the development of special sensors for their specific infrastructure. Based on this hypothesis, the methodology lists the sensor requirements for design to data analysis to accomplish this goal. In this case, the owner directed that the sensor would be low-cost and simple to fabricate, and that the data be related to the traffic activities. In this research, the authors engaged the owners and used their input for the design and development of the sampling rate, the accuracy of the data, energy needs for the long-term, and the locations of the sensors, prior to the sensor development. The infrastructure studied in this case is an aerial tramway, but the same methodology can be adopted for other applications. The steps of this methodology can serve as guidelines to develop similar applications extending the use of monitoring techniques to different structures [4,16,24], owners and operations that have a direct impact on the efficiency and sustainability of their activities [29].

To better accomplish early, low-cost, owner-centered sustainable monitoring, we determined a sustainable methodology: (1) deployment time duration, (2) hardware, (3) power supply, (4) software, (5) energy consumption, (6) data management and (7) device testing to make sure that the system was ready to deploy. In this methodology, we worked simultaneously with the owners of the tramway to decide what would be the minimum number of sensors needed to collect enough information.

### 2.1. Long-Term Requirements for LEWIS

LEWIS 4 had to be autonomously powered in order to work for long periods of time. One of the main obstacles when designing any kind of long-term deployment electronic device is how to provide continuous energy. Small single-use alkaline batteries are not a sustainable source of energy. However, for low consumption sensors, these are generally enough and can power the sensor over several months [33]. However, when the device has to perform tasks uninterruptedly for days, as in our case, the energy consumption increases substantially and energy supply becomes a major challenge that needs to be solved.

Solar energy is a sustainable and efficient way to obtain power from the surrounding environment. Solar energy harvesting using solar cells is a reliable source of energy for devices which, for any reason, cannot depend on the electrical grid [19]. In the past, researchers developed solar harvesting for WSSs that was able to generate 100 mW/cm$^2$ of power in a cheap, simple and reliable way [25].

Energy harvesting solutions needed to be designed to collect energy according to the expected climate conditions of Albuquerque, New Mexico. The shortest day has 9 h of sunlight and 15 h of darkness. We designed the power supply system based on the shortest day scenario.

Finally, we wanted to design and test the solar panel efficiency towards the sun in terms of sky cloudiness, solar panel angle and the position of the sun during the light hours of the day [34].

### 2.2. LEWIS 4 Hardware Components

Figure 1 shows the hardware layout of LEWIS 4 grouped in the following three clusters, followed by their purpose: (1) power system to supply the energy needed for long deployments; (2) microcontroller to process the code; and (3) sensor elements to collect and manage the data.

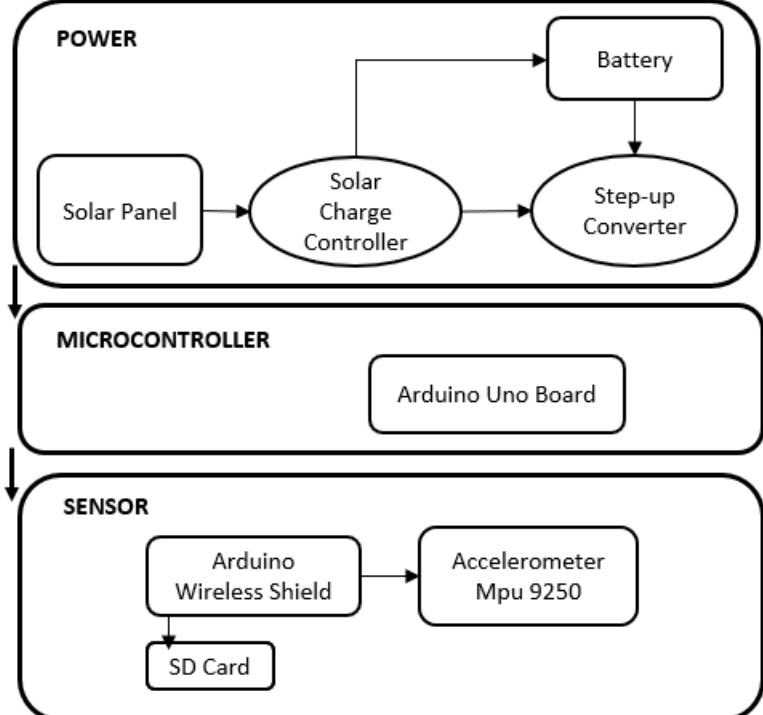

**Figure 1.** LEWIS 4 hardware layout.

### 2.3. Power Supply Hardware

We designed the power supply for long-term requirements and the worst adverse environment requirements expected at the Tramway location. The following sections describe each one of the components related to the power supply of LEWIS 4.

Solar Panel: Solar panels come defined both by voltage and power. These two parameters define the maximum output current that can be obtained in perfect conditions: perfectly oriented panel perpendicular to the sun lights; no clouds, panel surface perfectly clean; and maximum sun power; which is in general obtained in the mid hours of each day.

We defined a worst-case design scenario of 40% solar efficiency during 9 h of sunlight to define the requirements of the solar panel. In this worst condition scenario, the supply system must be able to keep the sensor working for 24 h so it can be charged daily. The solar panel to supply any long-term deployed sensor project must be able to charge the battery during the day so that the battery can provide a continuous power supply to the sensor during the night hours. Following this requirement, LEWIS 4 can rely on an autonomous stable source of energy provided by a solar panel.

We chose a 6 volt, 6 watt solar panel for LEWIS 4. The research team found that this solar panel, even working at 48% of its maximum capacity during the shortest day of Albuquerque, can still provide enough current to keep the system working autonomously.

Technical properties: nominal voltage, 6 V; peak power, 6.15 watts; dimensions, 220 mm × 175 mm (8.7″ × 6.9″); weight 225 g; cell type, monocrystalline; cell efficiency: 19%.

Battery: Although computational requirements have increased drastically over the last 30 years, battery technologies have seen little advance in the same period [35]. The most common are the nickel cadmium (Ni-Cd), nickel metal hydride (Ni-MH), sealed lead acid (SLA), lithium ion (Li-Ion), and polymer lithium ion (Polymer-Li or Li-poly) [36]. Table 2 describes the main characteristics of the most common battery cell types in the market in terms of materials.

**Table 2.** Properties of commonly used batteries [36].

| Cell Type | Ni-Cd | Ni-MH | SLA | Li-Ion | Polymer-Li |
|---|---|---|---|---|---|
| Energy density (Wh/kg) | 50 | 75 | 30 | 100 | 175 |
| Life cycle (charges-discharges) | 1500 | 500 | 200–300 | 300–700 | 600 |
| Self-discharge (charge % at time) | 60% 4 months | 15 % 1 month | 60% 24 months | 40% 5 months | 8% 1 month |
| Nominal voltage (V) | 1.25 | 1.25 | 2 | 3.6 | 2.7 |

For implementation outdoors with autonomous energy harvesting systems for long-term projects the battery needs to be rechargeable and must have a low self-discharge rate [37]. Another aspect to consider is the discharge curve [38]. The discharge specification relates the voltage of the battery to the capacity [39]. In general, this relation is not linear and it is conditioned by the discharge rate, the temperature and the battery material. Lithium-ion and lithium-polymer-ion battery voltages tend to drop fast when they are at maximum capacity, then their voltage stabilizes around their nominal value and finally the voltage drops fast again when they are approaching the maximum discharge zone [40].

We chose a lithium-ion 3.7 volt battery with a wide range of capabilities, which is easy to acquire and has a low-cost value, for LEWIS 4. This is a common battery available for a wide range of capacities and is easy to acquire.

Technical properties: 3 × 2200 mAh lithium-ion 18,650-sized cells in parallel; dimensions, 69 mm × 54 mm × 18 mm (2.7″ × 2.1″ × 0.71″); weight, 155 g (5 oz); max charging rate, 1.65 A; max continuous discharge rate, 3.3 A.

Solar charger: One of the common challenges when designing a new solar charger is that the amount of current provided by the solar panel is not stable. It can occur on any given day that an occasional shadow produces a sudden decline of the current amplitude. This may result in battery damage. To avoid the potential variation in the amount of input and the battery damage, a solar charge controller was added between the battery and the

solar panel, consisting of a step-down power converter. The power converter changes the 6 V of the solar panel to the 3.7 V of the battery, aided by a capacitor to stabilize the current being the input to the battery.

Step-up converter: Additionally, the system requires a step-up converter to feed the micro-controller. It steps-up the voltage from the 3.7 V of the battery to the 5 V that the Arduino USB port requires. The step-up converter has a USB port that simplifies the connection to the microcontroller.

Microcontroller: A microcontroller is an electronic component that users program to perform automatic operations such as digital data collection, postprocessing, or communication, among others. Microcontrollers are in general composed of a processor, memory, and external connections to allow additional functions. LEWIS 4 uses an Arduino Uno Rev3 microcontroller board. Arduino Uno is based on the ATmega328P, has 14 digital input/output pins (of which 6 can be used as PWM outputs), 6 analog inputs, a 16 MHz quartz crystal, a USB connection, power jack, ICSP header and a reset button. Additionally, it can be powered using alternating current (AC) adapters, most USB chargers or the USB port on a computer.

Sensor Elements: LEWIS 4 hardware uses Arduino-based boards. Arduino is an open-source programming environment, making it easy to find add-ons and a wide range of alternatives and options for sensors. Other advantages are its flexibility, user-friendly interface and smooth communication between devices [41].

Sensor Board: The LEWIS 4 sensing board consists of a GY-91 board with an embedded MPU9250 module. The sensor is equipped with an accelerometer with a dynamically selectable measurement range that varies from ±2 g to ±16 g in all three axes, x, y and z.

SD Card Shield: We selected an SD card shield that housed two main functions: (1) headers enabling an antenna connection for user communication; and (2) an SD card slot for data harvesting.

SD Card 16 GB: LEWIS 4 can generate data at a rate of 14 MBytes/h, meaning that a 16 GigaByte SD card has enough capacity to record up to 48 days. The SD card is subjected to high demands of data printing due to the high frequency of data being saved from the three channels. This can cause a potential ceasing of operations and should be overcome. We formatted the SD card using Fat 32 and 4096 byte allocation. The LEWIS 4 code uses the Sdfat library for logging data on the SD card at very high rates [42].

The figures below show the hardware elements as described above grouping them in sensor measuring elements, Figure 2. And the power supply elements, Figure 3.

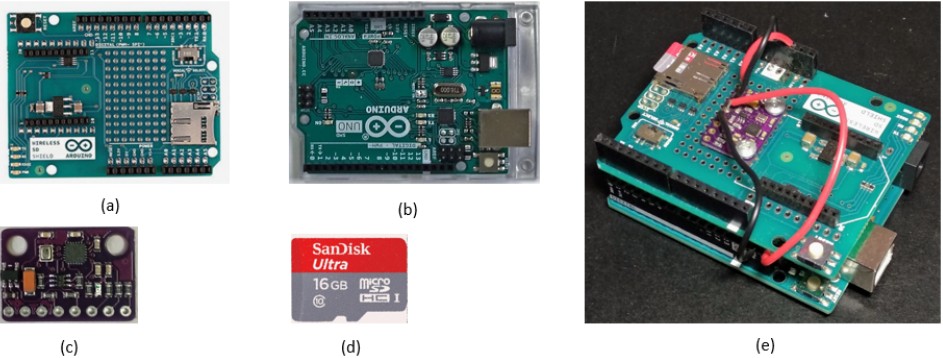

**Figure 2.** Sensor measuring elements: (**a**) Arduino wireless shield; (**b**) Arduino Uno; (**c**) GY-91 board (MPU9250); (**d**) 16 GB SD card; (**e**) assembled sensor.

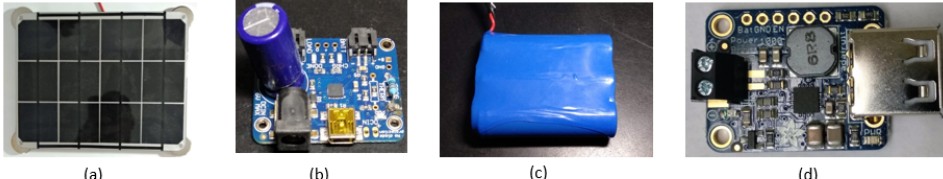

(a)　　　　　　　　(b)　　　　　　　　(c)　　　　　　　　(d)

**Figure 3.** Power supply system elements: (**a**) solar panel output: 6 V 6 W 1 A; (**b**) solar charger controller input: variable 6 V panel–1 A (added 2 KΩ; resistance), output: stable (3.7 V–1 A); (**c**) battery 3.7 V–6600 mAh; (**d**) step-up converter input: 3.7 V–1 A; output: 5 V–1 A.

### 2.4. LEWIS 4 Software

As addressed previously, LEWIS 4 is designed and built on an Arduino platform in order to enable low-cost implementation by owners. Arduino offers its own integrated development environment (IDE) and uses C++ coding language. Open-source Arduino libraries have been used and modifications have been introduced to enhance its characteristics and adapt it to the long-term deployment requirements. The software modifications make the system reliable and avoid errors while the sensor is recording. Two main modifications were made in order to achieve the goals for field implementation and use by stakeholders: energy consumption reduction and enabling a user-friendly interface. LEWIS 4 records data at a sampling rate of 250 Hz. LEWIS 4 also modifies the maximum number of blocks that a file can reach per file to 150,000 blocks, which is equivalent to 5.5 h of continuous recording. LEWIS 4 starts recording a new file automatically when the previous one reaches its maximum size. LEWIS 4 also starts recording automatically when the sensor is started. By default, the sensor, when powered, offers some options and waits for the user to type a command to start any of its functions. This modification introduced in LEWIS 4 leads the sensor to automatically start recording right after powering the sensor. This makes it simple to deploy and also means that it will automatically recover from a power loss, which was brought up as a potential need by stakeholders. Finally, LEWIS 4 automatically deletes incomplete files recognized by the code as temporary files that can cause the sensor to stop recording. In general, all these modifications are solutions to the different problems that researchers found during the development of the LEWIS 4 in conjunction with the Tramway deployment. Figure 4 shows the software architecture diagram of LEWIS 4.

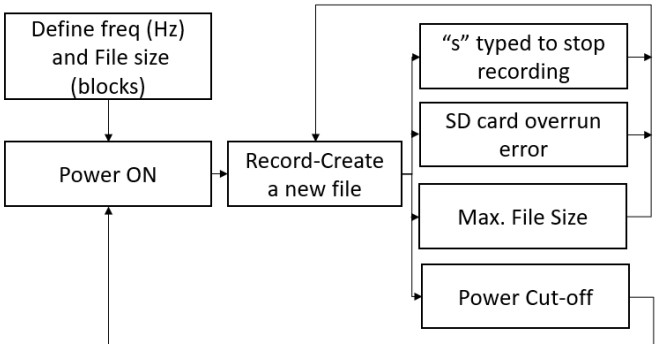

**Figure 4.** LEWIS 4 monitoring architecture.

### 2.5. Energy Consumption

Wired sensors operate with a stable power source but wireless systems depend on autonomous power supply systems that must be able to supply the required energy in a stable manner. Although there are several research studies about operating systems, data management and sensor software, it is accepted that energy consumption is still a limiting factor [43].

We designed and developed a sustainable, reliable energy supply system for LEWIS 4 to work independently of any external energy source. We determined the energy requirements of 138 mAmps at 5 V, when powered and recording. We used these values to estimate the energy requirements for LEWIS 4 and its field implementation on the Tramway.

The battery capacity (reservoir) must be enough to supply a stable output of energy (to the sensor) of 4320 mAh daily, with a variable daily input (solar panel) of $\alpha \, mAmps \times \beta \, hours$, where $\alpha$ is the power output of the solar panel and $\beta$ depends on the solar exposure hours during the day.

### 2.6. Field Data Pre-Processing

Data management is a problem when collecting data continuously at 250 Hz for several days. LEWIS 4 saves data in a binary format generating (.bin) files. These .bin files are convenient for logging data at high frequencies. However, binary format cannot be used to plot or analyze the data. The files must be converted to a readable format, in this case comma-separated value (.csv) files. Csv files are especially convenient for working with large amounts of data. In order to provide stakeholders with a workable format, LEWIS 4 was coded to provide .csv files.

LEWIS 4 converts (.bin) files to (.csv) in an acceptable amount of time with software written by us, i.e., in approximately 10–20 s [44]. The data processing includes (1) deleting the data that are not necessary; (2) setting a threshold to keep only the interesting values; and (3) converting the raw values given by the sensor into engineering units with the formulas below.

$$\frac{Sensor\,acc\,data}{\frac{16}{2^{15}}} * 9.81 = Acc(\mathrm{m/s^2}) \tag{1}$$

$$\frac{Sensor\,gyro\,data}{\frac{2000}{2^{15}}} = Gyro(\mathrm{degrees/s}) \tag{2}$$

where "Sensor acc data" denotes the data directly recorded by the sensors divided by the calibration factor of the sensor.

### 2.7. Testing and Validation

LEWIS 4 was tested in a controlled environment to check that the solar energy supply system enables the sensor to record uninterruptedly and to save the data correctly. Figure 5 shows the battery fluctuations through the day together with the amount of current provided to charge it by the solar panel during the 48 h experiment to test and validate the energy consumption design for LEWIS 4 prior to the field deployment. The sensor was set to record continuously during 48 h with a sunlight exposure of 9 h a day, which was determined to be the shortest daylight duration. We recorded the battery voltage every 5 h. Fortunately, the data showed that LEWIS 4 battery levels fluctuated as expected. The solar panel provided the current for charging the batteries at a fast rate during the two days of testing. The batteries were charged with energy to supply the sensor until the next day. In summary, the sensor recorded continuously for 48 h and the charge–discharge rate confirms that the system is able to supply energy at a sufficient rate to keep the sensor working uninterruptedly in good conditions in the field. In Figure 5 the blue line (continuous line) represents the battery voltage, which can be used to estimate its remaining capacity relating to the characteristic discharge curve of the battery. In Figure 5 the orange line (dashed line) shows the current provided by the solar panel. We found with this experiment that in the test conditions the solar panel worked at a 93% efficiency providing an average of 0.93 amps over the maximum output that is 1 amp. This current is enough to charge the batteries during the day so that they can keep the sensors working overnight. Note that during this test was performed in May in Albuquerque (NM) and the sky was clear so sun exposure was excellent.

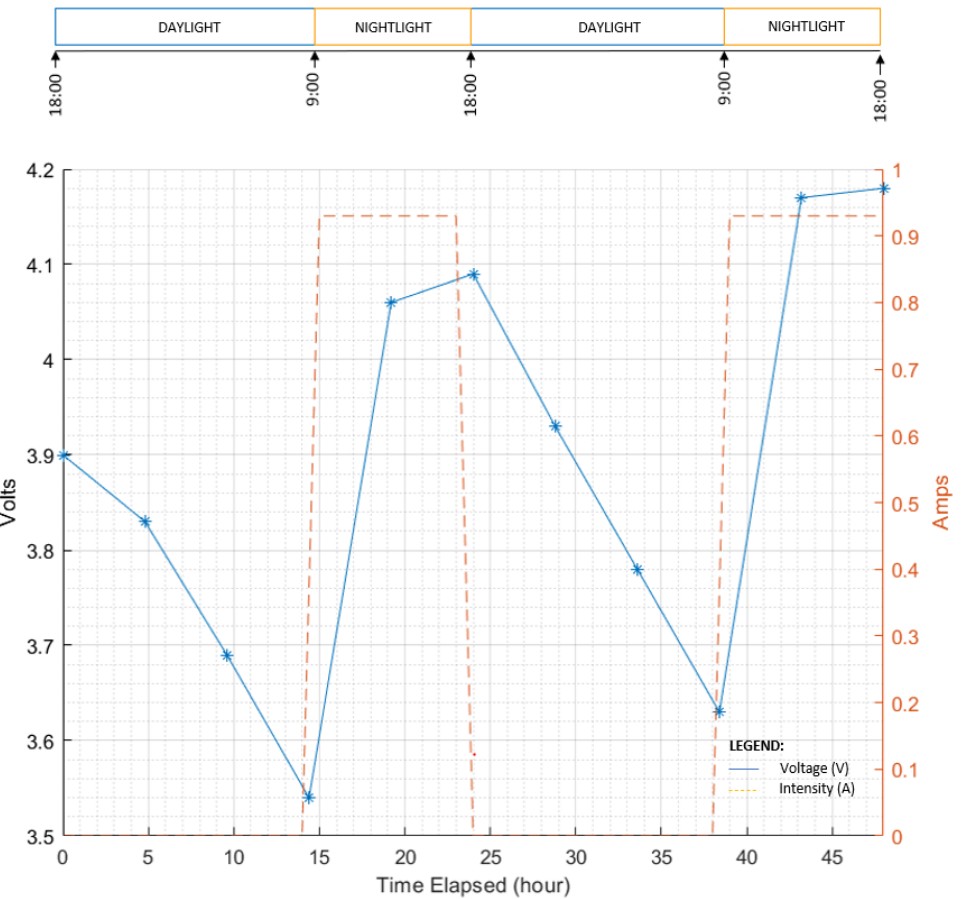

**Figure 5.** Solar energy harvesting validation during 48 h prior to field deployment.

## 3. Field Deployment of LEWIS 4 Sensors in the Sandia Peak Aerial Tramway

After validating the sensors in controlled outdoor environments on campus, we planned the field deployment for the Sandia Peak Tramway infrastructure and its operations. The long-term monitoring capability enables the monitoring of the aerial Tramway by adding the two sensors in locations with difficult access. The sensors enable collection of data at those remote locations autonomously. According to the Sandia Peak Tramway, access to permanent information for the operations is currently only possible by allowing their operators to observe the car and tower operations on their own. Therefore, Sandia Peak Tramway is interested in collecting this data remotely. Additionally, Sandia Peak Tramway emphasized that objective data can be useful to their staff to compare operations and inspections across time.

### 3.1. Tramway Structure Description

Figure 6 offers a general view of the tramway structure and its location on the mountain while Figure 7 shows descriptive plan and elevation views of the Sandia Peak Tramway. Opened in 1966 after 24 months of construction, the Sandia Peak Aerial Tramway is an iconic structure of the city of Albuquerque and one of the most known attractions in New Mexico. It is located on the northeast edge of Albuquerque and connects the city side foothills of the Sandia Mountains with its crest. The Tramway ascends vertically 3819 feet (1164 m) on a 2.7 mile (4.34 km) cable in just 15 min. The cable is supported by two towers; the cable span between Tower 2 and the upper base is the third longest Tramway cable span in the world spanning 7720 feet (2353 m). This is also the longest Tramway in the United States. When the experiment was run, the tram cars operated daily from 9 AM to 9 PM.

The Sandia Peak Tramway is a bi-cable double reversible aerial Tramway. The two cars have capacities of 55 people each and they are attached to the same haul cable. The weight of one of the cars when running downhill helps pulling the uphill tram car. The maximum speed is 24 feet per second, equivalent to 12 mph or 20 km/h. Each car weighs 8000 pounds (3628.7 kg) when empty and has a maximum payload of 10,000 pounds (4535.92 kg). They are made mostly of aluminum to lighten the weight.

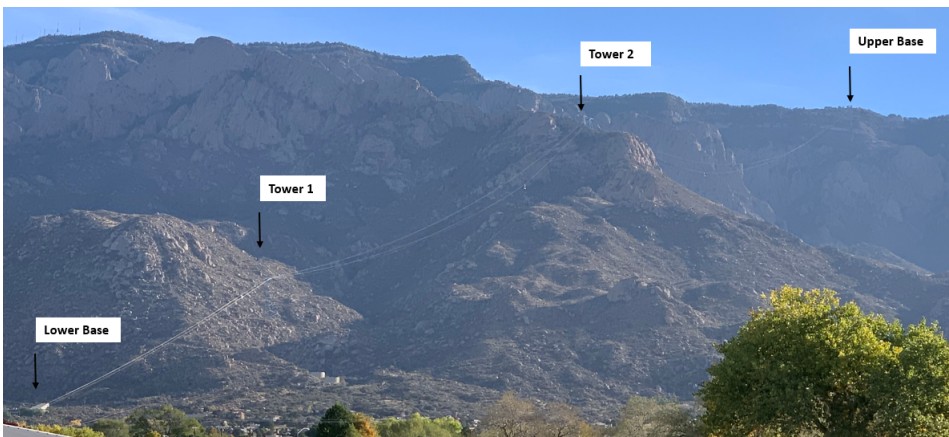

**Figure 6.** General view of the Tramway and the Sandia Mountains from Albuquerque.

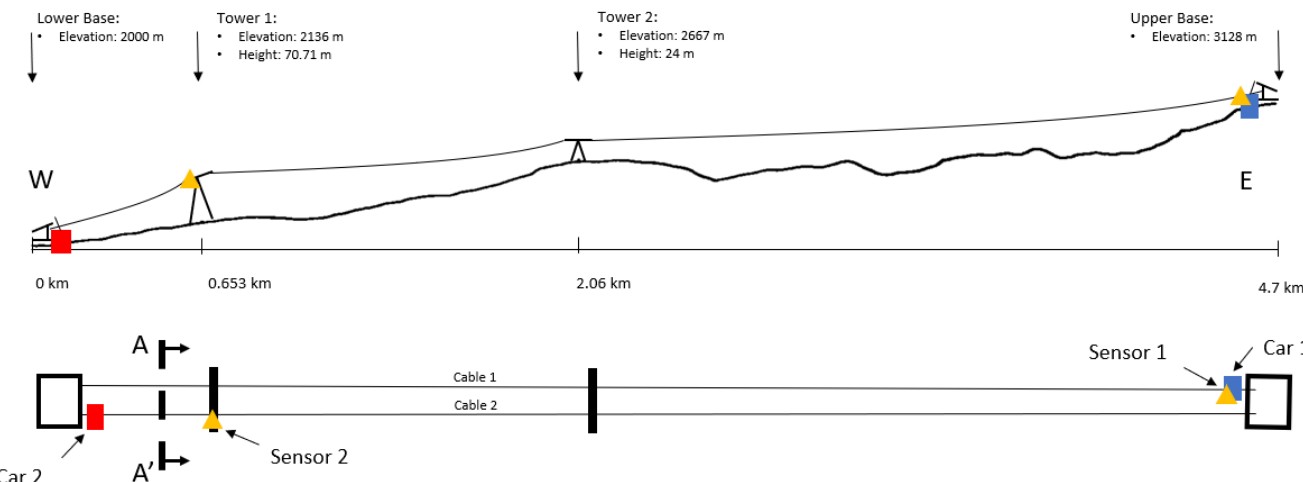

**Figure 7.** Descriptive figure of the structure and location of the sensors. This figure shows no operations with the two cars on their bases.

### 3.2. Instrumentation

The research team installed for the Sandia Peak Tramway two LEWIS 4 sensors to monitor operations. One LEWIS 4 was placed on the roof of the car (Sensor 1), while the second one was placed on the top of Tower 1 (Sensor 2). The sensors were designed to be fixed with magnets to the steel structures. Figure 8 shows the details of the LEWIS 4 installation for both locations. The details and location of LEWIS 4 were discussed and decided with Sandia Peak Tramway experts in operations.

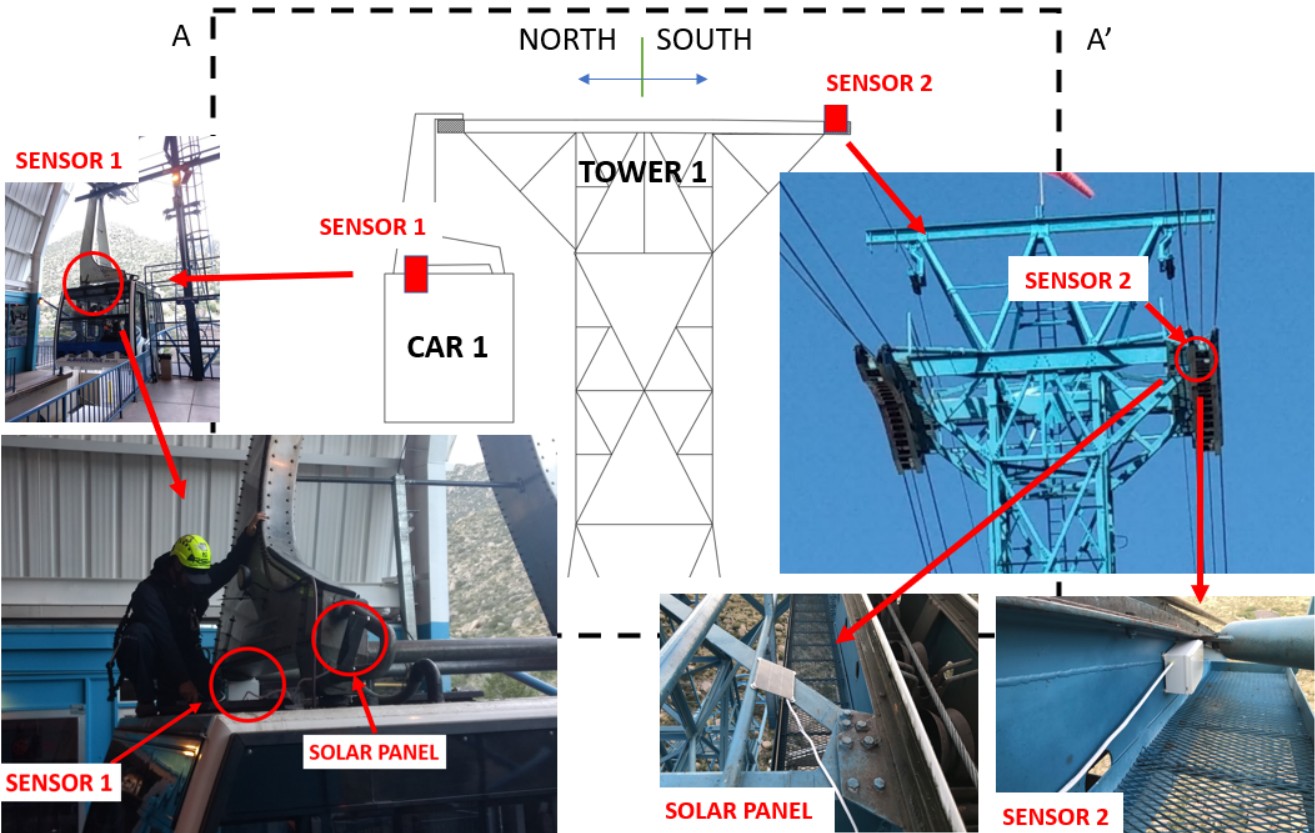

**Figure 8.** Cross section A-A' (Figure 7). Sensor location description as viewed looking from west to east or from lower base to top.

LEWIS 4 number 1 was installed on Car 1. The best location to install the car sensor was the steel arm that holds the passenger box connected to the top of the car. This also allowed maximum exposure to sunlight during the day.

LEWIS 4 number 2 was installed on Tower 1. The research team placed the tower sensor on the tip of one beam that supports the tram cables in order to collect information about the pull forces on the structure.

### 3.3. Sensor Deployment and Retrieval

On the day of the deployment we started Sensors 1 and 2 at 6:47 AM prior to installation. Sensor 1 was installed a couple of minutes after being started at 7:05 AM on the top of Car 2 at the lower base, making sure that the solar panel received the most sunlight from the south. Sensor 2 was installed at 11:47 AM on the top of Tower 1. The sensors recorded operations uninterruptedly day and night and were removed after three days during regular maintenance activities in order not to disturb Sandia Peak Tramway operations.

### 4. Monitoring and Data Analysis

The acceleration values collected by the two sensors allowed us to estimate the position and speed of the cars. Multiple flights were recorded. The data collected by both LEWIS 4 sensors were curated and benchmarked using the flight records for the same period of time provided by Sandia Peak Tramway. For the purpose of automation of operations management, we used one flight. We also found information about how the structure responds to the dynamic loads of the cars on movement and how the friction of the cable with the towers is an action that affects the structure, and Sandia Peak Tramway wanted to take this into account for their internal maintenance and management. Figure 9 shows the data collected by both sensors during this flight and it serves as a good example to illustrate the value of automatic monitoring of operations using LEWIS 4. According to the

log-book provided by the Tramway managers this was flight number eleven of the day, which departed at 12:24 AM. Car 1 with Sensor 1 recording started its descent on Cable 1 from the uphill base. Sensor 2 was on Tower 1, placed on the Cable 2 support beam on the south side.

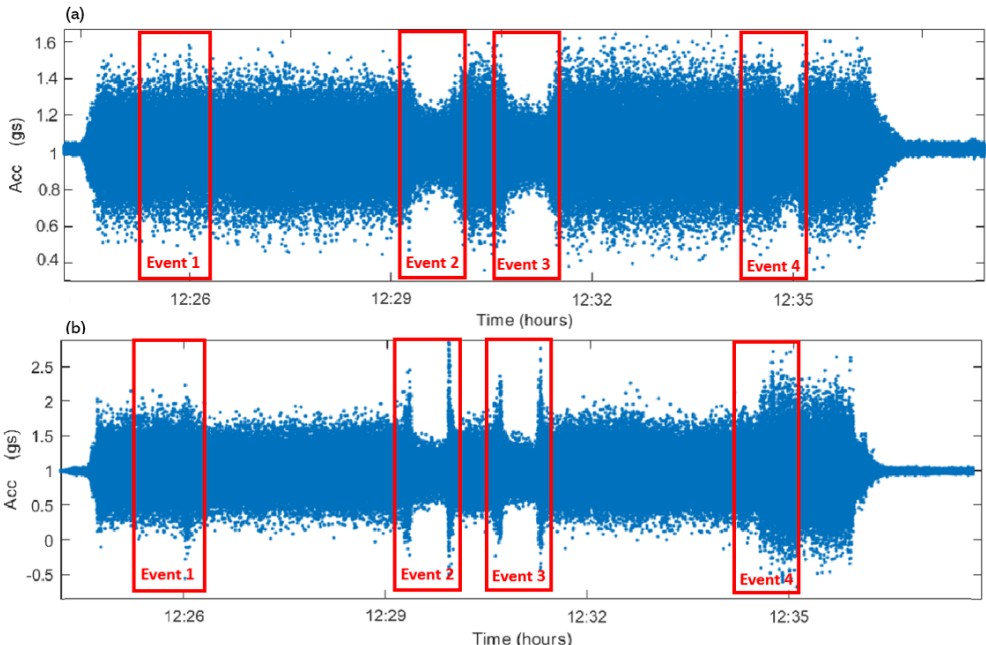

**Figure 9.** Vertical axis acceleration values recorded during flight 11 by the Car 1 (**a**) and the Tower 1 (**b**) sensors.

*Data Analysis*

This section summarizes the monitoring of one flight using data from Car 1 descending from the upper base and Tower 1 to illustrate the value of automated recording of events in relation with operations of interest to the Sandia Peak Tramway. We found four remarkable events that quantify patterns of operations of interest from every flight. These patterns allow us and the owners to estimate the position and speed of the Tramway cars automatically. The owners and managers can use these data to generate an automatic log of operations. Furthermore, they can quantify changes in responses between flights, days and seasons by using the amplitudes of the events. We found the following information of interest associated with each event. Car 2 rides uphill on Cable 2 and Car 1 rides downhill on Cable 1. The acceleration of both sensors is on the vertical axis, which was the most significant. The data collected is shown in Figure 9. Figures 10–13 show Sandia Peak Tramway Event 1, Event 2, Event 3 and Event 4, respectively. The following sections outline the automation of operations inspections for each event.

The results provided the following insight: the highest acceleration values on the tower appear when the cable accelerates and decelerates, modifying its speed. This interesting finding was the result of a discussion with the owners on the governing structural concerns in their operations and also in coordination with the discussion of the data obtained in the field, identifying an unexpected source of dynamic load on the tramway tower that is actually related to maintenance and operations. More specifically, the owners identified that quantifying these changes in velocity can inform them on operations and maintenance as the cable maintenance is of top importance to their operations. More specifically, the lubrication on the cable bearing is a maintenance activity that needs to be carried out frequently and information about the activity can benefit the operations and safety, with low cost and limited effort in terms of data analysis and number of data points, which is of high value to the owners. These findings demonstrate that a simple, low-cost, small number of sensors co-designed and co-developed with the owner can

engage the investigation with the owner and hence discover applications that are useful for maintenance management, and would not have been discussed otherwise. In the opinion of the authors, the selection of a simple effort in number, cost, and short-term deployment in collaboration with owners can inform early areas of interest and value from them, since (1) their time and resources are limited and (2) they are interested in simple concepts that can be related with specific activities. Additionally, based on the success of this shared effort, the authors propose that similar efforts are conducted with owners of other infrastructure such as railroads, parks and outdoor facilities such as ski resorts or similar private infrastructure. Their direct feedback and early engagement to simple deployments can inform the further development of monitoring on larger scales. This incremental collaboration is more sustainable and based on our experience, generates early findings and results.

It was a remarkable finding for us to determine the relation between vibration and speed, especially on the tower. The highest acceleration values on the tower appear when the cable accelerates and deaccelerates when modifying its speed. This is an interesting finding and it was not expected by us. This experiment helped us to identify an unexpected source of dynamic load on the tramway tower. These conclusions were communicated to the structure owners who commented that the lubrication on the cable bearing is a maintenance activity that needs to be carried out frequently. In our opinion, these findings prove that not only traffic monitoring, but also the sensors, are useful for maintenance management.

**Event 1** (Figure 10). The values recorded for Car 1 (blue) are not altered in this event. Car 2 (red) rides uphill and crosses Tower 1 (green). The acceleration recorded for the tower sensor decreases. This decrease is caused by the reduction of the cable speed when Car 2 is crossing Tower 1. Once the car crosses Tower 1, the acceleration of the tower stays constant until Event 2.

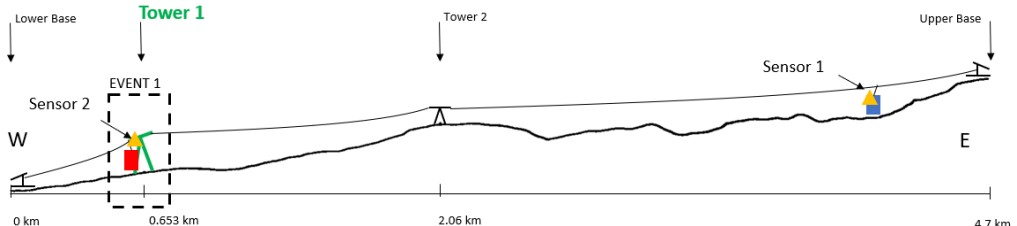

**Figure 10.** Event 1. Car 2 crosses through Tower 1.

**Event 2** (Figure 11). Car 2 rides uphill and crosses Tower 2. This event is detected by both sensors. The values recorded for Car 1 (Sensor 1) decrease considerably in this event. The data from Sensor 2 (Tower 1) show a remarkable acceleration amplitude peak, continued by a decrease in amplitude, and a much higher peak. This signature shows a severe reduction in the speed of the cable when the car is crossing the beam of Tower 2. The two peaks show the moment when the cable abruptly changes its speed. The two cars go at a much lower speed through Tower 2 than they do through Tower 1 and the cable, when decelerating and accelerating, generates higher vibrations on Tower 1.

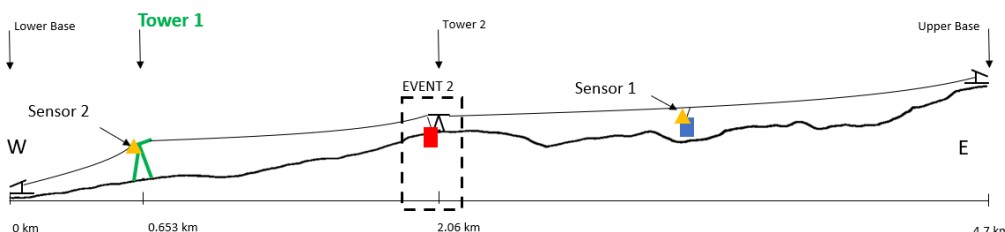

**Figure 11.** Event 2. Car 2 crosses through Tower 2.

**Event 3** (Figure 12). Car 1 rides downhill and crosses Tower 2. The data observed show a similar response to the data in Event 2. It is worth mentioning that, similar to Event 2, Car 1 (Sensor 1) data decrease in amplitude. Based on the data, the acceleration of the car depends mostly on the speed of the cable.

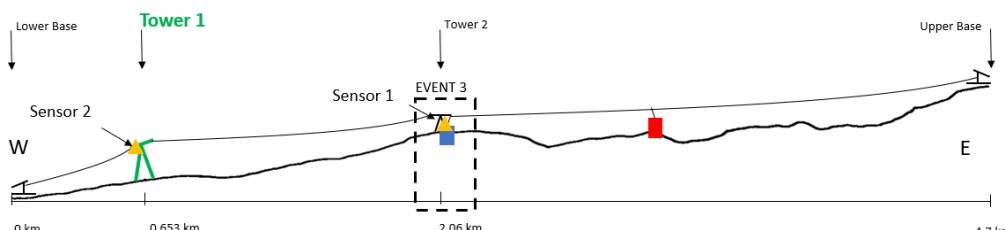

**Figure 12.** Event 3. Car 1 crosses through Tower 2.

**Event 4** (Figure 13). Car 1 rides downhill and crosses Tower 1. The acceleration recorded by Sensor 1 (Car 1) decreases. The acceleration recorded by Sensor 2 (tower) increases. The increase in acceleration amplitude in Sensor 2 is generated because Car 1 is now carried from the portion of the cable between the lower base and Tower 1.

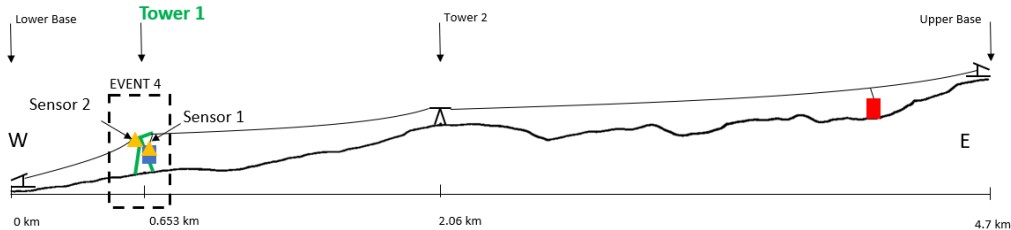

**Figure 13.** Event 4. Car 1 crosses through Tower 1.

This analysis demonstrates that the data assist owners in quantifying relations between speed, the position of their operations and accelerations. By examining the acceleration data collected by just two sensors, the location of the two cars can be found, and at what speed and in which direction they are moving, which is a useful tool to record traffic activities on the Tramway.

Some conclusions that can also be obtained from the analysis are summarized below to show the additional information provided by LEWIS 4 that is of interest to operators:

Firstly, the acceleration values recorded on the tower vary in a range of $\pm 2$ g while the ones on the car just $\pm 0.8$ g. This indicates that the Tramway cars perform well in controlled accelerations, and also reinforces the value of collecting Tower accelerations to inform the owners about the performance of their entire system from both operations and maintenance perspectives.

Secondly, the largest vibrations recorded by Sensor 2 (Tower 1) are caused by the cable friction reducing and increasing cable speed when the cars are passing by Tower 2. This is seen in Events 2 and 3 in Figures 11 and 12. The crossing of Tower 2 is of critical importance to Tramway owners, and it can be used for automatic monitoring and recording of such events. The data match the expectations of the owners.

Finally, the sensor located in the car does not record strong vibrations when crossing through the towers. Contrarily, car vibrations are due to the movement of the car at higher or lower cable speed. This was not expected by the research, but the Tramway owners expected this result and could use this information in the future.

## 5. Conclusions

Two acceleration sensors strategically located in one of the cars and one of the towers of the Sandia Peak Aerial Tramway were enough to collect data that let researchers determine the position and the speed of the cars along the tramway line. In addition, we found

that the acceleration data collected by the tower sensor provides information about the interaction between the cable and the bearings on the tower. This interaction and the pulling forces generated by the friction on the system are a critical maintenance issue on the tower. The two LEWIS 4 acceleration sensors, designed in collaboration with the structure owners, have proved to be a cost-efficient and sustainable solution to integrate automatic monitoring into the system and provide operators with some data to assist their activities.

Infrastructure owners are going to be reluctant to invest in complex monitoring systems unless they are offered an ad-hoc solution specifically designed to address their needs, which provides them with useful and valuable data. This research advances in this direction. After finding that acceleration data provide information about position, speed and cable-tower interaction, further research must be carried out to integrate the systems with the operator controls and develop an interface that lets the operator visualize and obtain assistance from these data in real time.

Future research efforts must be directed towards developing automated data management to automatically process analyze and interpret the data. Researchers must also work to interconnect multiple sensors as part of a network in order to analyze and correlate data from different parts of the structure.

**Author Contributions:** Concept and development: R.C.H. and F.M.; test design: R.C.H. and F.M.; analysis of results: R.C.H.; writing—original draft preparation: R.C.H.; writing—review and editing: R.C.H. and F.M.; supervision: J.A.L.G. All authors have read and agreed to the published version of the manuscript.

**Funding:** This research was supported in part by the Department of Civil, Construction and Environmental Engineering of the University of New Mexico; the School of Civil Engineering of the University of Castilla-La Mancha; the Transportation Consortium of South-Central States (TRANSET); the US Department of Transportation (USDOT), Projects No. 17STUNM02 and 18STUNM03; the New Mexico Consortium Grant Award No. A19-0260-002, and the project BIA2017-86811-C2-1-R founded with FEDER founds.

**Data Availability Statement:** Some or all data, models or code that support the findings of this study are available from the corresponding author upon reasonable request.

**Acknowledgments:** The authors would like to give thanks for the resources from the Center for Advanced Research and Computing (CARC) of the University of New Mexico. The authors wish to thank Michael Donovan and Joe Puliaco (Sandia Peak Tramway) and George Boyden (Sandia Peak Tramway retiree) for their assistance and suggestions for the installation, development and advancement of low-cost wireless sensors. Finally, Jack Hanson and Xinxing Yuan assisted the field deployment and their support is appreciated.

**Conflicts of Interest:** All authors declare no conflicts of interest.

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
