# Peer review of "Aerial Tramway Sustainable Monitoring with an Outdoor Low-Cost Efficient Wireless Intelligent Sensor"

_sustainability, doi:10.3390/su13116340_

Round 1

Reviewer 1 Report

I do not see a scientific purpose in this article. This is an engineering case report.

Author Response

Please find attached as a word file

Reviewer 2 Report

  1. The introduction should highlight and clearly identify the research issues.
  2. The introduction ends with Table 1, but lacks analysis or highlights of what makes the data relevant.
  3. Chapter 2 is more reminiscent of literature analysis than the methodology on which research will be conducted, and so on. The exact research methodology is not defined
  4. The text itself lacks a greater contribution from the authors, i.e. there is a lack of insight / analysis of the authors of the article, as well as opinions on the data analyzed and obtained.
  5. The conclusions do not name any further research.

Reviewer 3 Report

A new Low cost Efficient Wireless Intelligent Sensor LEWIS 4 is developed for long-term monitoring of civil infrastructures. LEWIS 4 is a sustainable self-powered sensor that uses a solar energy harvesting power supply system. The main novelty of the new generation of LEWIS is the long-term capability and autonomous data acquisition. The problem addressed in this paper is genuine and the scope of the sensor application and design are well-explained in the paper. However, there are some issues that need to be addressed in the paper prior to acceptance for publication which are listed as below:

  • Using self-powered sensors for structural health monitoring is a hot topic in the literature. The authors are recommended to cite some of the state-of-the-art papers related to this topic such as https://doi.org/10.1016/j.engstruct.2019.109619 in which new data analysis techniques are developed for structural health monitoring.
  • The effect of environmental and operational variations on the data collected from the sensor should be considered, at least as a future work. To this end, the authors are recommended to cite some of the recently published articles on this topic such as https://doi.org/10.1016/j.ymssp.2021.107847 and https://doi.org/10.1016/j.ymssp.2017.10.013.
  • At the beginning of Line 67: correct “use” to “uses”.
  • Figure 1 is not well-organised. For instant, the connection between Power supply system, Microcontroller, and the Sensor is not well depicted. Please consider revising.
  • Line 155: correct “Whereas” to “Although”.
  • Line 161: correct “energy harvest systems” to “energy harvesting systems”.
  • Line 181: Should “s” in “step-Down” be capital as well?
  • In Figure 4, it appears that some lines are duplicated. Please consider revising.
  • Figure 5 is very trivial; the authors are recommended to present more complex relationships in a figure as such. Therefore, please either consider revising or deleting this figure.
  • Equations 1 to 3 are not well written. The authors are recommended to write up formulas in a more organised manner.
  • Line 274: place a comma after the term “Fortunately”.
  • Line 280-284: Consider adding the type of the lines after mentioning their colors, i.e. blue (continuous line) and orange (dashed line). Also, please add a legend to Figure 6.

Round 2

Reviewer 2 Report

  1. Adjust the title of chapter 2 by introducing more clarity because it is not clear whose methodology? sensors? or what?
  2. If possible, move Table 2 closer to the reference text
  3. Specify the title of Section 2.6
  4. It is recommended to expand the comments on the results obtained to Figure 5

Author Response

please find attached in PDF for your convenience, thank you

the authors

This manuscript is a resubmission of an earlier submission. The following is a list of the peer review reports and author responses from that submission.

Round 1

Reviewer 1 Report

The title and summary do not specify what physical quantities are monitored by the sensor. This is weird.
No scientific purpose indicated. No practical purpose was indicated.

The work does not indicate the "superiority" of the proposed solution over other alternatives.

The work is written very well, but it is promotional and advertising work. It is not an academic work.

There is no scientific analysis of the measurement results.

Figure 9 is obvious. It is a reproductive element in this work.

Fig. 10 is illegible and in this form is not suitable for scientific analysis (which is not included in the rest of the work).

The presented conclusions are not adequate to the content of the article.
The conclusions are very general, not justified by the content of the article.
Work and conclusions cannot be scientifically confirmed.

Reviewer 2 Report

GENERAL COMMENTS:

In the document “Aerial tramway sustainable monitoring with an Outdoor Low-cost Efficient Wireless Intelligent Sensor (LEWIS4)” the authors present the development and application of a low-cost monitoring system for aerial trams. It is an autonomous system, powered by solar energy, and which has hardware is based on the Arduino computational platform equipped with sensors (accelerometers) to collect acceleration data to estimate the position and speed of cars. The pilot project was implemented at the Sandia Peak Air Tramway in Albuquerque, New Mexico, where two LEWIS4 modules were installed. Data were collected, presented and discussed with information on the number of flights, time, speed and position of the cars. I think it is a consistent project, since the authors have been working on it for some time, and the version presented in this document is the fourth. The document meets, in my view, the necessary requirements for a scientific document, as it presents a good structure and good writing. Below I leave some specific comments related to the parts that make up the document, individually.

SPECIFIC COMMENTS:

TITLE:

The title is of an appropriate size, with 13 words, (Aerial tramway sustainable monitoring with an Outdoor Low-cost Efficient Wireless Intelligent Sensor (LEWIS4)). It must be taken into account that the title is the first component to be read in an article and, therefore, is the most important phrase of the same. It should not be forgotten that the reader can select our article for reading from the assigned title and, therefore, it should reflect its content, be concise and include the most relevant terms of the objective of the work. Regarding the writing, the title should answer two fundamental questions: What was done? I think it answers both questions. However, I see it as unnecessary to include the acronym LEWIS4.

RESUME:

The "summary" is very important due to its significant use in electronic databases. The "summary" presented in the document is of an appropriate size (200 words), distributed in 9 sentences and presenting an average of 22.3 words per sentence / sentence. I think the number of words per sentence is over the limit. Studies indicate that the sentences should not contain more than 15 or 16 words, in order to make reading more pleasant and without interruptions.

In relation to construction, I think that the abstract presents structural problems and, in my opinion, presents the necessary items for this part of the document. Namely: “What has been studied” (Introduction); “How was the study done” (Materials and Methods); "What was found" (Results) and "What it means" (Conclusion). I think that the text presented does not fully address these requirements, for example, it could make the materials and methods used and the conclusions clearer. Another pertinent observation is that the abstract must be written in the past tense, except for the last paragraph or the concluding sentence. This does not occur in part of the text. So, I am of the opinion that the abstract needs to be rewritten.

KEYWORDS:

The authors present 7 (Aerial tramway; automation; sustainable sensor; long-term sensing; low-cost sensing; solar energy; transportation sustainability). I think the number is quite high, I would recommend a number of 3 to 5 words. I think the word Arduino should be included, as it would attract attention, since it is a very widespread platform. I always recommend consulting a thesaurus, because the keywords are the most important in the article and provide the correct cataloging of the article once published. Keyword suggestions can be found in the magazines themselves or on other specific websites, for example:

  • http://ieeeauthorcenter.ieee.org/wp-content/uploads/taxonomy_v101.pdf; https://trends.google.com.br/trends/explore;
  • https://ubersuggest.io ;
  • http://keywordtool.io;
  • https://eric.ed.gov/?ti=all; among others.

INTRODUCTION:

This section aims to explain why the study is necessary and why it should be published. It should summarize the study's justification and should interest the editor and the reader. The introduction must clearly establish the nature and scope of the problem studied. I think that the section presented though seeks to present the general themes that cover the problem, the justification and the objectives. It is possible to visualize the importance and scope of the study and the limitations of the study.

The Introduction section presented is thus structured:

  1. Introduction

1.1. Motivation

1.2. Long-term Requirements for LEWIS

1.3. LEWIS 4 Hardware Components

1.4. Power Supply Hardware:

1.5. LEWIS 4 software

1.6. Energy consumption

1.7. Data Management

1.8. Test and validation

I think that the structure presents does not fulfill the desirable for this section. I leave a suggestion, of sequential structure, to rewrite this part:

  • Enumeration of the general themes that cover the problem (theory).
  • Review of the background of the problem.
  • Definition of the research problem (question).
  • Statement and location of the variables (forecast and result) to be considered in relation to the problem.
  • Formulation of study objectives.
  • Importance and scope of the study.
  • Study limitations.
  • Document roadmap.

It is observed that practically all the items are contemplated in the document, however, they need to be adjusted.

Items 1.2 to 1.8 should be in other sections. For example, in a section that deals with the materials and methods used. I do not understand that topics such as: requirements analysis, hardware specifications, data collection and management and testing and validation should be in the Introduction section.

METHODOLOGY or MATERIALS AND METHODS.

This section should describe how the research was carried out. That is, it must present in an organized and precise manner how each of the proposed objectives will be achieved. The section dealing with the methodology should describe in an orderly manner and in chronological sequence what was done (not what was found), in order to be possible to reproduce / replicate the work developed.

The document presented does not have a specific section. I am of the opinion that the authors should include a section on Material and Methods. I would suggest that you include a graphic element that could show the research phases and also briefly comment on each phase. Once this is done, I think it will become easier to understand the results obtained, as they need to demonstrate consistency with the applied methodology.

RESULTS AND DISCUSSION

The epilogue of a survey is to show the results. These must be in agreement with the formulated objectives and be consistent with the proposed methodology. I think that the weakness found in the methodology section had a negative impact on the presentation of the results, in terms of their understanding. Although the results presented are significant. The data collected is very rich and certainly supports the research carried out. Perhaps it was interesting that the discussion could explain the implications of the results found in relation to other works by other authors. I think this section of the document is very rich, but it should be more in line with a methodology section.

FINAL CONSIDERATIONS or CONCLUSIONS

People usually read the abstract, the introduction and the conclusion. The content of the conclusion should start by emphasizing the main message and the main result that supports it. The data reported in the results section must be mentioned (without repeating the figures), their importance must be weighed and the implications described. If possible, the interpretation and comparison of the results in relation to the main message. The new and important aspects of the study and the conclusions that are derived from them must also be emphasized. Authors should take into account that the discussion is an opportunity to explain the results and help the reader to understand the study, specifically the new and useful knowledge.

I think that, although the authors worked on the section, it still owed a little.

REFERENCES

The authors provide a list of 36 references that are all cited in the document. I think the number of references is suitable for documents with the proposed profile. It drew attention to the fact that 63.3% of the references are dated before 2015, that is, more than five years old, which, in terms of an article that involves technology, is a little worrying. Four undated references were also found.

GRAPHIC ELEMENTS

Figures, tables and charts are intended to communicate information visually and quickly. The authors present 14 figures and 2 tables, I think it is a very good number and they helped to understand the project and the data. I leave it as an observation to be careful with the dimensions of some figures.